# Monitoring of "7-20" Rainstorm Damage in the Zhengzhou Road Network Using Heterogeneous SAR Images

**Siyi Li [1], Guowang Jin [2],*** and Jiahao Li [2]

1 School of Surveying and Land Information Engineering, Henan Polytechnic University, Jiaozuo 454003, China
2 Institute of Geospatial Information, Information Engineering University, Zhengzhou 450001, China
* Correspondence: guowang_jin@163.com

**Abstract:** Flooding is one of the most frequently occurring meteorological disasters nowadays, and its occurrence can cause significant socio-economic losses. Aiming at the problem that traditional optical remote sensing makes it difficult to monitor floods, this paper designs a scheme to jointly extract the scope of the affected area by using heterogeneous satellite-based SAR images acquired at different times within the flood period. This paper takes the "7-20" rainstorm in Zhengzhou City as an example and uses two kinds of heterogeneous SAR images, Sentinel-1A and GF-3, to extract the flooding damage in the main urban area. In addition, combinations with the vector data of the Gaode road network were performed to further monitor and analyze the "7-20" rainstorm damage in the main urban area of Zhengzhou. The results showed that the main urban area of Zhengzhou City was affected by the "7-20" rainstorm. The roads in the main urban area of Zhengzhou were seriously affected, and the total length of the affected roads reached 1324.63 km. The monitoring scheme for flooding road network damage using Sentinel-1A and GF-3 heterogeneous SAR images has certain feasibility.

**Keywords:** rainstorm monitoring; threshold method; road damage area; heterogeneous SAR

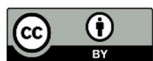

## 1. Introduction

Extremely heavy rainfall phenomena frequently occur in China. According to data released by China's Ministry of Emergency Management, floods affected a total of 59.01 million people in 2021. These floods caused direct economic losses of 245.89 billion yuan, accounting for 73.62 percent of natural disasters' total direct economic losses. After the flooding disaster, quickly and effectively identifying the storm inundation area and the damage to the features is an essential guide for the relevant departments to grasp the disaster data and improve the emergency rescue and disaster prevention and mitigation capabilities.

In recent years, the rapid development of remote sensing technology has made it widely used in pre-disaster prediction and post-disaster rescue [1]. The methods to achieve water body extraction by remote sensing flood dynamic monitoring can be classified into three types: threshold, object-oriented, and deep learning [2]. The threshold method, also known as model classification, classifies the images by analyzing the spectral characteristic curves of water bodies. Select a suitable band to construct a model and an appropriate classification threshold to obtain a binary map of water bodies and non-water bodies [3]. The main methods of image segmentation based on thresholding are the histogram bimodal method, Otsu maximum interclass variance method, EM algorithm (Expectation-Maximum), water body index method, etc [4]. Mason et al. monitored flooding in urban and rural areas in the UK using a threshold method and change monitoring [5]. Martinis et al. fully automated near real-time flood detection by fuzzy logic and threshold methods for Germany, Thailand, Albania, and Montenegro [6]. Mcfeeters used TM images

in green and near-infrared bands to construct the Normalized Difference Water Index (NDWI) to extract water body information within the city limits, which can better suppress vegetation information [7]. The NDWI can hide the vegetation information and suppress the influence of soil, buildings, and shadows to highlight the water body information. This method is simple, fast, efficient, and widely used in water body extraction. In addition, it can achieve better classification results in plain areas with slight variations in topographic relief [8,9].

The object-oriented method takes the image object as the primary processing unit. It makes homogeneous image elements into objects of different sizes by various segmentation algorithms, thus realizing the extraction of image information by an object as a unit [10]. With the increasing spatial resolution of remote sensing images, the classification methods to achieve target feature extraction using the images' rich spectral and complex texture features are becoming increasingly sophisticated [11]. Yu et al. introduced multiple SAR image texture features for water body information extraction and compared them with the traditional classification method to improve water body extraction accuracy [12]. Foroughnia uses supervised and unsupervised classification methods, combined with multispectral and SAR data, to assess the precision and accuracy of flood extraction [13].

Deep learning has been widely used in remote sensing water extraction with its unique advantages, such as powerful feature representation and automatic feature learning from data through deep neural network structures [14], commonly used deep learning image segmentation algorithms FCN (full convolutional networks) [15] and U-Net networks [16]. Li et al. combined multi-temporal TerraSAR-X data and interferometric coherence as training samples to propose an active self-learning time-integrated convolutional neural network framework (A-SL CNN) [17]. Xu presents a novel Synthetic Aperture Radar (SAR) image change detection method that integrates effective image preprocessing and Convolutional Neural Network (CNN) classification [18]. Results show that the proposed method has higher accuracy in comparison with traditional change-detection methods. Wang et al. applied the constructed full convolutional neural network model for water body extraction experiments. The results showed that the fully convolutional neural network model is more automated, better applicable, and has higher extraction accuracy than the traditional threshold method for water body extraction [19].

Since floods are mostly accompanied by various types of cloudy and rainy weather extremes, traditional optical remote sensing images cannot obtain ground information in a timely and accurate manner. SAR satellites can acquire ground information around the clock and in all-weather due to their wavelength characteristics [20]. More and more methods are based on SAR images to monitor the changes in water bodies, which has become an indispensable data source in flood emergency monitoring [21]. However, most current studies mainly focus on improving the accuracy of water body extent extraction, and relatively little research has been conducted on the damage to road networks in flooded areas. The urban road network is dense in form and complex in structure, carrying various types of traffic flows, which is the key to maintaining the city's regular operation. Therefore, it is essential to identify the inundated location and damage to the road network under extreme weather conditions for post-disaster relief. In the event of flooding, due to the number of satellite constellations and revisit cycles, a specific type of satellite often does not meet the longtime series monitoring requirements for flooding, which also becomes an important problem for flood rescue and post-disaster assessment.

To quantitatively monitor the road damage in the "7-20" rainstorm in Zhengzhou, this paper designs a scheme to extract the road network damage in the flood disaster by combining the heterogeneous SAR data and the road network data before, during, and after the rainstorm. At the same time, a time-series monitoring of the disaster process of the "7-20" mega rainstorm is carried out. The length of flooded roads was extracted by the SAR image threshold segmentation method and GIS spatial analysis method to provide data support for quantitative road damage assessment.

## 2. Study Area and Data

### 2.1. Study Area

Zhengzhou is a megacity and one of the major economic centers in central China, the capital city of Henan Province, and an important railroad and highway transportation hub in China. 12.6 million people live in Zhengzhou City, the first in the province, according to the seventh national census in 2020. Zhengzhou is bordered by the Yellow River to the north and has 124 large and small rivers in its territory, spanning two significant basins: the Yellow River and the Huai River. The Yellow River basin includes parts of Gongyi and Shangdi, with an area of 2011.8 square kilometers, accounting for 27% of the city's total area. The Huai River basin includes all of Xinzheng, Zhongyuan District, Erqi District, Guancheng District, and parts of Xinmi, Jinshui District, and Huiji District, with an area of 5499.5 square kilometers, accounting for 73% of the city's total area. The city has 124 rivers of various sizes, with 29 rivers with larger watershed areas (≥100 square kilometers), including 6 in the Yellow River basin and 23 in the Huai River basin. The rivers crossing the border are the Yellow River and the Ilo River.

In July 2021, under the guidance of the airflow of the Pacific subtropical high pressure, a large amount of water vapor was continuously transported from the sea to the land, which was influenced by the topography to collect rain within Henan Province. From 18:00 on 18 July to 0:00 on 21 July, Zhengzhou received heavy rainfall, with a cumulative average precipitation of 449 mm. The single-day downpour broke the 60-year historical record since the establishment of the Zhengzhou weather station in 1951. This exceeded the regional flood control and drainage capacity [22]. The Investigation Report of the "720" Very Heavy Rainstorm Disaster in Zhengzhou states that the heavy rainstorm caused extensive flooding in urban and rural areas, severe flooding in urban streets and depressions, short-lived flooding in rivers and reservoirs, and direct economic losses of 40.9 billion yuan. This study selected the road network in the main urban area of Zhengzhou City (Jinshui District, Huiji District, Zhongyuan District, Erqi District, and Guancheng Huizu District) for monitoring, and the specific study area is shown in Figure 1.

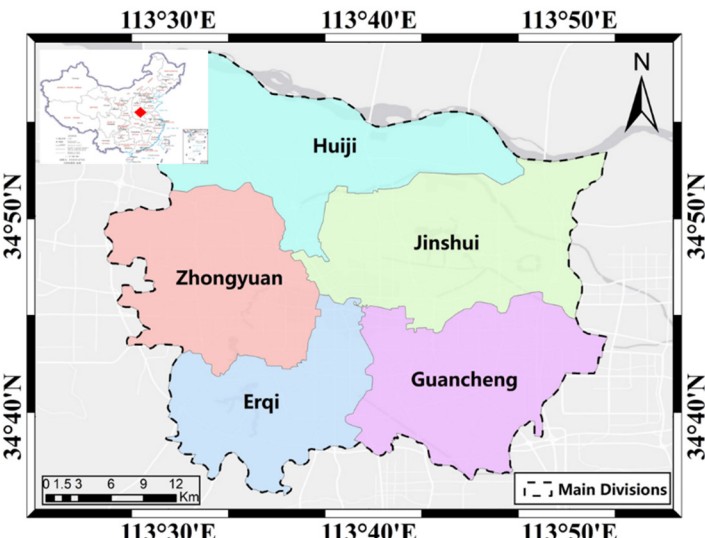

**Figure 1. The** main urban area of Zhengzhou. The black dotted line represents the boundary of the main urban area of Zhengzhou. The red diamond shows the location of Zhengzhou on the map of China.

### 2.2. Different SAR Datasets

Sentinel-1A is the first satellite developed by the European Commission and the European Space Agency for the Copernicus Global Earth Observation Project, which was

launched in April 2014 with C-band imaging. Sentinel-1A contains four operating modes: SM, IW, EW, and WV, including an interferometric wide-field mode with a resolution of 5 m × 20 m and an amplitude of 250 km and a revisit period of 12 days. Sentinel-1A SAR data are freely available, providing rich data support for global scholars to monitor the global land and coastal zone.

The GF-3 satellite, the first Chinese high-resolution SAR remote sensing satellite with 1 m spatial resolution, was launched on August 10, 2016, using C-band imaging and containing 12 imaging modes. Depending on the imaging modes, GF-3 can provide multi-polarized SAR images with 1 m to 500 m resolution and 10 km to 650 km width to achieve global monitoring of ocean and land resources [23]. The imaging mode used in this paper is fine striping with HV polarization and 10 m azimuthal resolution of the image.

Due to the satellite revisit cycle at the time of the "7-20" rainstorm disaster, this paper combines two kinds of heterogeneous SAR satellites, Sentinel-1A and GF-3, as well as the observation data of the "7-20" rainstorm in Zhengzhou. The Sentinel-1A VH SAR image was imaged on 15 July 2021. The two views of GF-3 SAR images were imaged on 20 and 22 July 2021, respectively, and HV polarization was used. The specific parameters of the heterogeneous SAR images with different phases are detailed in Table 1. Each SAR image is shown in Figure 2.

**Table 1.** Experimental data of this study.

| Data Type | Imaging Time | Band | Imaging Mode | Polarization | Resolution/m |
|---|---|---|---|---|---|
| Sentinel-1A | 15 July 2021 10:20 | C | IW | VH | 5 × 20 |
| GF-3 | 20 July 2021 22:28 | C | FSII | HV | 10 × 10 |
| GF-3 | 22 July 2021 10:38 | C | FSII | HV | 10 × 10 |

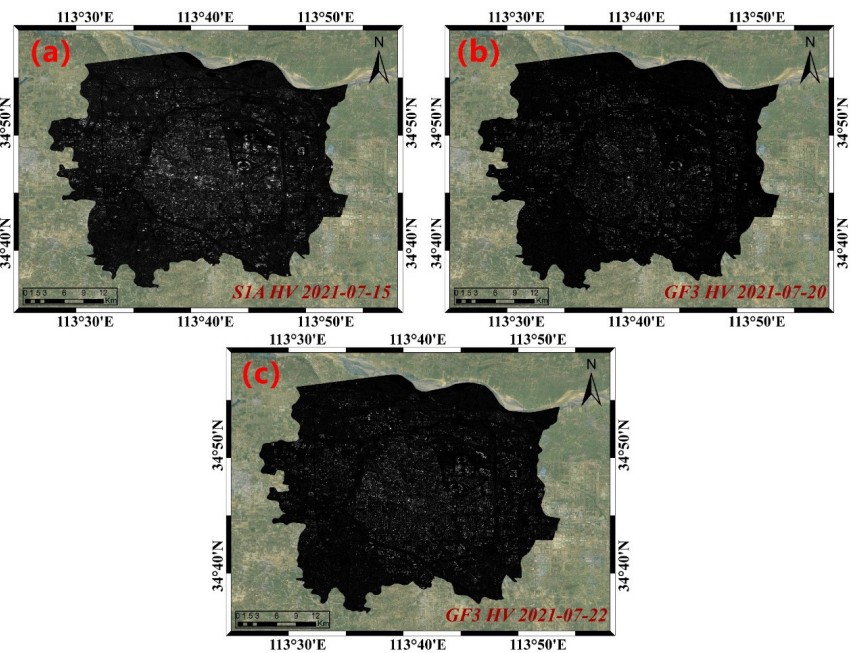

**Figure 2.** Heterologous SAR images in different phases ((**a**) is Sentinel-1A's 15 July 2021 VH-polarized SAR image (before the disaster); (**b**) is GF-3's 20 July 2021 HV-polarized SAR image (at the time of the disaster); (**c**) is GF-3's 22 July 2021 HV-polarized SAR image (after the disaster); the bottom image is Google's remote sensing image of Zhengzhou City).

### 3. Methods

This paper used VH-polarized Sentinel-1A images on 15 July 2021, and HV-polarized GF-3 images on 20 and 22 July. We selected the dB values of typical water bodies in different SAR images through image alignment, multi-viewing, filtering, geocoding, radiometric calibration, and other processing. Then, we applied the threshold segmentation method to extract water bodies in different SAR images to generate before, during, and after the rainstorm. The binary maps of water bodies and non-water bodies are generated before, during, and after the rainstorm. After acquiring the changes in water bodies, this paper combines the vector data of the Gaudet road network to count the length of affected roads using the GIS spatial analysis method. Figure 3 shows the road damage monitoring flow chart in Zhengzhou City during the "7-20" rainstorm using heterogeneous SAR images.

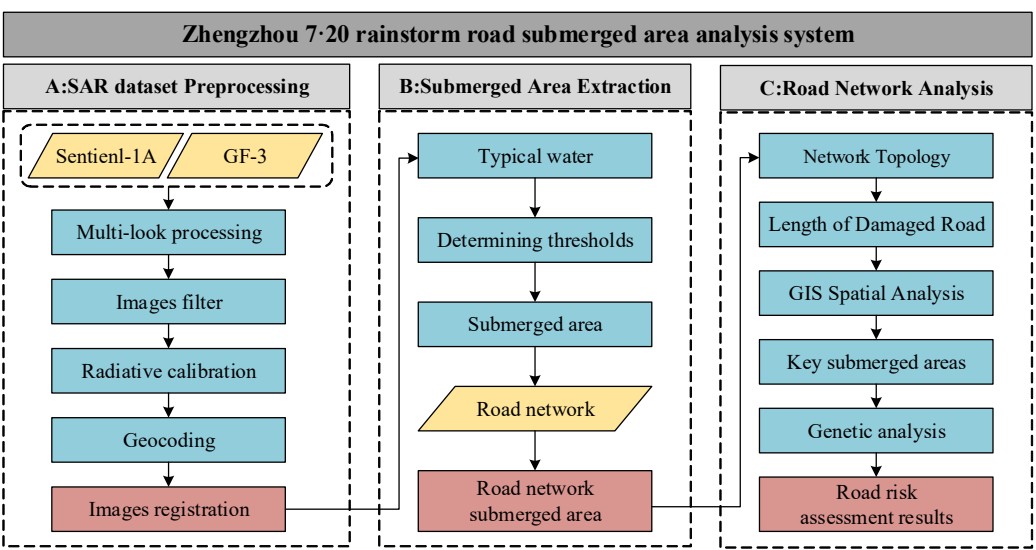

**Figure 3.** Flow chart of heterogeneous SAR images for road damage monitoring in Zhengzhou City during the "7-20" rainstorm.

### 3.1. SAR Dataset Processing

The pre-processing of the heterogeneous SAR images of Sentinel-1A and GF-3 mainly includes the steps of image alignment, multi-view, filtering, radiation correction, etc. Figure 4 shows the processing of the GF-3 image on 20 July.

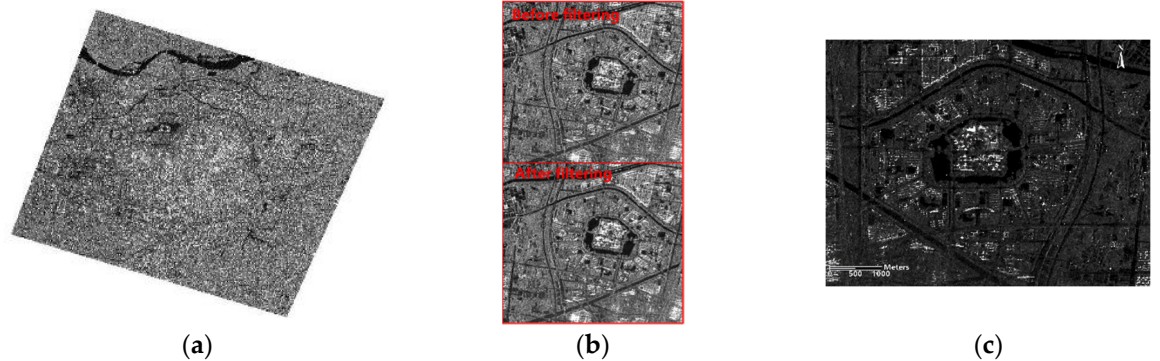

(**a**)          (**b**)          (**c**)

**Figure 4.** GF-3 SAR image on 20 July (drawing mode is Square Root). (**a**) Multi-looking. (**b**) Filtering. (**c**) Geocoding and Radiometric Calibration.

(1) Multi-looking processing: The Sentinel-1A SLC image is multi-view processed with a view number ratio of 4 × 1, and the GF-3 SLC image is multi-view processed with a view number ratio of 2 × 2 to attenuate the influence of coherent speckle noise information on the image quality.

(2) SAR image filtering: In this paper, the enhanced Lee filtering method is used for SAR image filtering, which effectively overcomes the shortcomings of the traditional Lee filtering method for non-homogeneous areas with poor filtering effects and adopts the process of distinguishing the target distribution in the image area and divides the SAR image area into the homogeneous area, the non-homogeneous area, and the separation point target area. In this paper, the enhanced Lee filtering method with a window size of 5×5 is selected to filter SAR images, which can effectively maintain the edge information of radar images while suppressing noise information.

(3) Geocoding and radiometric calibration: Geometric and radiometric corrections are performed to eliminate distortions and obtain the backscatter coefficients of the images by using the 30 m resolution digital elevation data NASA SRTM released.

(4) Image registration: By selecting control points on the image, the three SAR images taken before, during, and after the rainstorm are registered and geometrically corrected.

The SAR system can obtain the power ratio of the measured emission and return pulses, and this ratio (backscattering) is projected to the slant range geometry to better compare SAR images' geometric and radiation characteristics under different SAR sensors and receiving modes. It is necessary to perform geometric and radiation calibration on the slant range SAR data and convert it into a geographic coordinate projection.

The range of geometric distortion in the SAR image is large, mainly caused by the change in terrain. Based on the given digital elevation model, the relationship between the three-dimensional coordinates of the ground point and the two-dimensional coordinates of the slant distance image is established by the distance-Doppler geometric model and the backward projection algorithm. The range-Doppler model is expressed as [20,21]:

$$R = S - P \tag{1}$$

$$f_D = \frac{2f_0(v_p - v_s)R_s}{c|R_s|} \tag{2}$$

where $R_s$ is the tilt range, $S$ and $P$ are the sensor and backscatter unit positions, $v_s$ and $v_p$ are the sensor and backscatter unit velocities, $f_0$ is the carrier frequency, $c$ is the speed of light, and $f_D$ is the processed Doppler frequency.

For better comparison of heterogeneous SAR image data, the radar data need to be radiometrically calibrated using the radar equation to achieve calibration of the radar backscatter coefficient. The radar backscatter coefficient refers to the radar reflectivity per unit area of the target in the incident direction, and the backscatter coefficient can also be regarded as a combination of three elements: unit cross-section, reflectivity, and directionality. The general form of the radar equation is:

$$P_r = \frac{P_t A^2}{4\pi\lambda^2 R^4}\sigma \tag{3}$$

where, $P_t$ is the transmitting power of the radar transmitter, $A$ is the effective scattering transceiver area, $R$ is the slope distance from the antenna phase center to the target point, $\lambda$ is the electromagnetic wave wavelength, and $\sigma$ is the radar scattering cross section.

### 3.2. Road Network Processing

To monitor the damage to roads in the main urban area caused by the July 20 rainstorm, the author obtained the road network data of the main urban area of Zhengzhou in 2021 using the Python web crawler tool. Road information includes ten categories of urban first-grade roads, urban second-grade roads, urban third-grade roads, urban fourth-grade roads, expressways, national roads, provincial roads, railways, county roads, and township roads in the main urban area of Zhengzhou. The processing of road network data mainly includes:

(1) Culling small roads. The rich variety of roads in the Gaode road network data contains some redundant roads with too short lengths. Fine roads less than 5 m were excluded from this study to reduce data redundancy.

$$RT = \{L|L > 5\} \tag{4}$$

where $RT$ is the set of road elements and L is the length of the road.

(2) Topology check. The original road network data is intricate and overloaded with details, so it is necessary to perform topological checks on the roads to avoid errors in the subsequent analysis. We check each line element for topological errors such as line segment self-intersection, line overlap, hanging points, and pseudo-nodes. Finally, we obtain the corrected road network data for the study area shown in Figure 5.

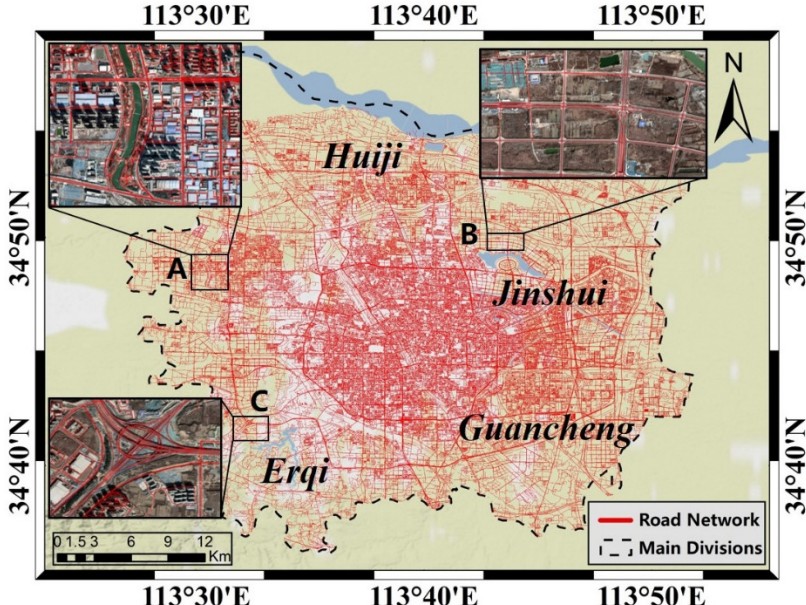

**Figure 5. The** road network and part of the corresponding Google image in the study area.

Three feature areas with different spatial distributions were selected to verify whether the processed road network data fit the actual road network. Their processed road network information was confirmed with the corresponding Google images. The three areas have different road network characteristics due to different geomorphological and spatial distribution characteristics. Area A is located near the intersection of Science Avenue and Xushui River Road, where the road network is dominated by Xushui River Road on both sides of the river, and the road network information of Science Avenue and Xushui River Road can be seen from the enlarged map of area A in Figure 5. The B area is located near the Longhubei subway station. The road network information in this area is relatively simple, and the comparison is also apparent. C area is located at the intersection of West Haoheng Road and West Fourth Ring Road, which is dominated by elevated bridges and has more accurate road network information. In summary, this paper's processed road network information is reliable after comparing it with Google images.

### 3.3. Water Body Extraction

Threshold segmentation uses the gray difference between the target and the background in the image to be extracted. It separates the target from the background by setting different thresholds to divide the pixel level into several classes. The general process is to determine whether a pixel point in an image belongs to the target or background region by judging whether each pixel point's feature attributes meet the threshold requirements, thus converting a grayscale image into a binary image.

$$g(x,y) = \begin{cases} 1, P(x,y) \leq T \\ 0, P(x,y) > T \end{cases} \tag{5}$$

where $g(x,y)$ is the image after thresholding, $P(x,y)$ denotes the image element value of the point $(x,y)$, the pixel marked as 1 corresponds to the target object, the pixel marked as 0 corresponds to the background, and $T$ is the target object segmentation threshold interval.

Determining the threshold value of water bodies on different images is key to this experiment. In this paper, we use manual empirical selection to superimpose the SAR images to be processed on Google Earth, find the obvious water body sample points such as water bodies and lakes, and identify the DN value of the water body image element. The DN value of the image element is the backscattering coefficient of the image element in dB. Figure 6 shows the backscattering coefficients of Sentinel-1A and GF-3 for two different SAR images of some typical water samples. Based on the analysis of the statistical characteristics of the sample points, the optimal interval where the water body thresholds are located is determined as [0, 0.001] for the Sentinel-1A image water body extraction threshold interval and [0, 0.009] for the GF-3 image water body extraction threshold interval in this experiment.

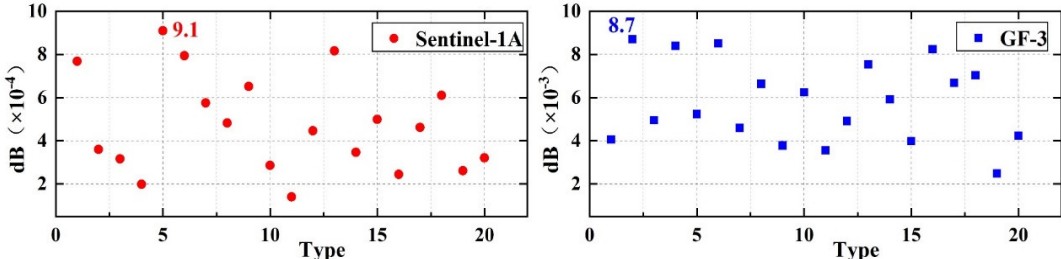

**Figure 6.** dB values of typical water bodies in different SAR images. The figure marks the highest dB value.

## 4. Results

### 4.1. Analysis of Regional Floods

Three radar images on 15 (before the rainstorm), 20 (during the rainstorm), and 22 (after the rainstorm) were extracted using threshold segmentation. The images before and after the rainstorm were superimposed to determine the extent of rainstorm inundation. This paper extracted and counted the flood disaster area time sequence changes in each main urban area in three periods after obtaining the flood disaster conditions in Zhengzhou. The change in water body area in each district (7.15, 7.20, and 7.22) is shown in Figure 7, and the detailed water body area is shown in Figure 8.

By observing the comparison of water body areas in the three time periods in Figures 7 and 8, the storm reached its maximum rainfall on 20 July, with huge fluctuations in water bodies in all five urban areas.

The inundated area of water bodies within Guancheng District increased the most, from 7897.10 m² before the rainstorm to 41,299.30 m², which was the most severely affected. As can be seen from the chronological changes in Figure 7, the village of Wuzhuang in the northeast of Guancheng District was severely affected, and the roads in that area were still not restored to normal in some of the affected areas until 22 July. The Changzhuang Reservoir in Zhongyuan District was also severely flooded, with an affected area of 27,174.9 m². Due to the continuous heavy precipitation and a large amount of water flowing upstream of the reservoir, the water level in the reservoir remained high. Affected by this, the water level of the river section in this area also rose to a certain extent. In addition, due to the reservoir spillway encroachment blockage and other circumstances, the water level in the Guojiazui reservoir in Erqi District on the 21 July rose significantly

to the maximum full overflow water depth of 0.5 m, the occurrence of the reservoir flooding became its 22 July retreat area of the least urban areas.

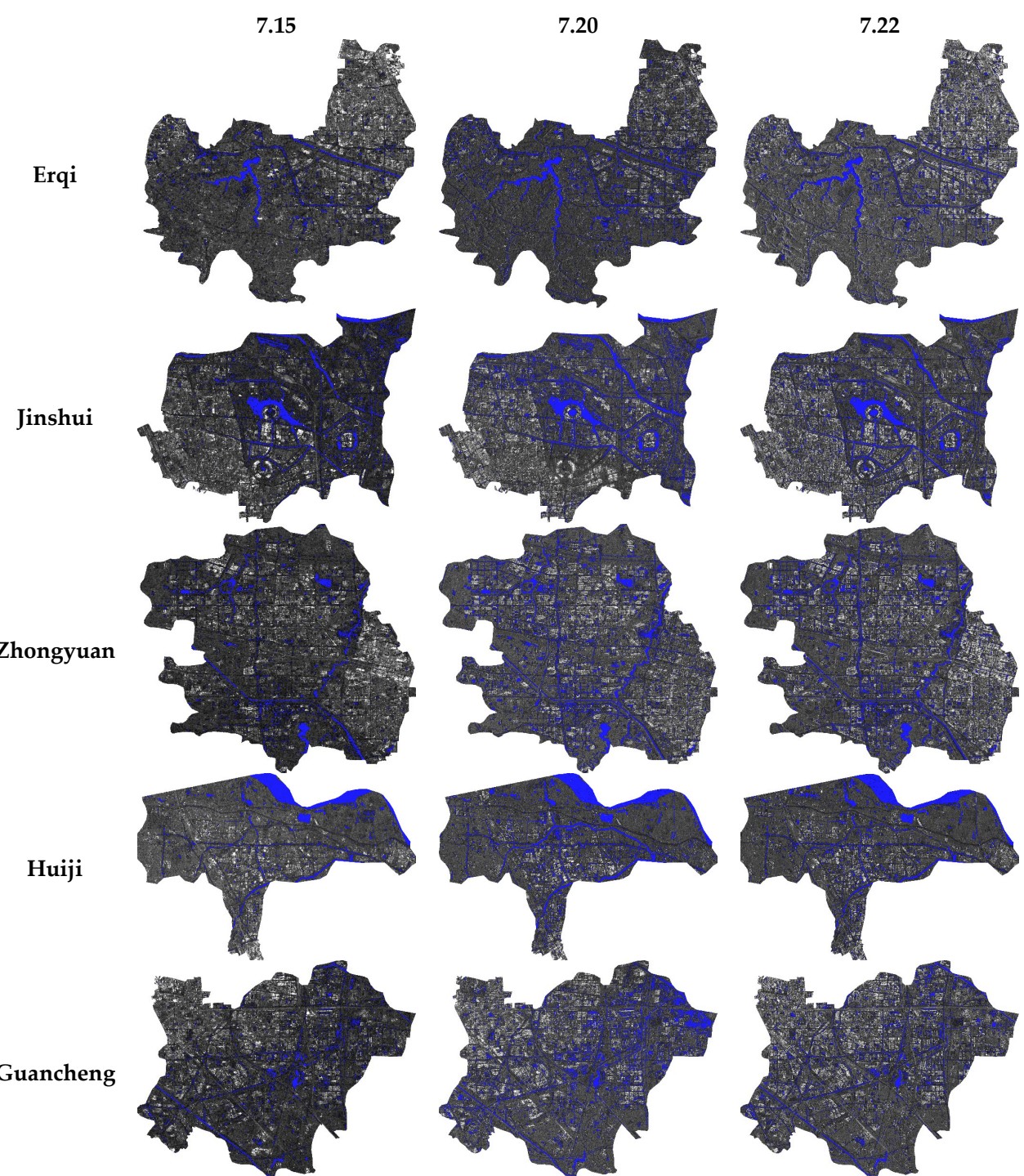

**Figure 7.** Water extraction results before, during, and after the rainstorm.

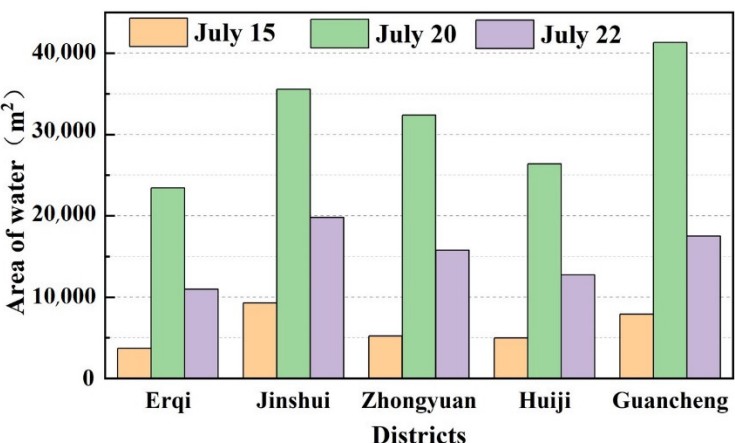

**Figure 8.** Variation of the water body area in each district of Zhengzhou.

Comparing the changes in water bodies in each district in Figure 8, Huiji District was the least affected by this extreme weather, which is inseparable from the unique natural environment within the region. Huiji District is located in the northern part of Zhengzhou City, and the terrain generally shows highs in the west and lows in the east. The Yellow River runs through the northern part of the district for 27 km, making it the richest ecological water system in Zhengzhou. In addition, the high forest cover in the district provides favorable conditions for stormwater infiltration and drainage. When heavy rainfall occurs in the area, the infiltration of water absorbed by the plant soil layer and the timely drainage of the river do not cause large areas of water to accumulate on the ground.

Overall, Zhengzhou's "7-20" rainstorm affected each of Zhengzhou's five main urban areas to varying degrees of disaster. As of 22:00 on 20 July, the area of water bodies in Guancheng District grew the fastest, with a total increase of 33,402.2 m², while the total area of water bodies in Zhongyuan District increased by 27,174.9 m². As of 10:00 on 22 July, the total area of water bodies in all urban areas of Zhengzhou City decreased compared to the total area of water bodies on 20 July. The total area of water bodies in all city's urban areas decreased compared to the 20th. Erqi District recovered the most slowly, with a recovery area of 12,461.1 m².

*4.2. Road Disaster Situation*

The inundation extent of the storm determined by the threshold segmentation method above was converted into vector surface layers, and the pre-processed road network data of the study area was overlaid, and the extracted inundated road network was shown in Figure 9 using the GIS spatial analysis method as shown in Equation (6). With the help of spatial analysis tools, the road network conditions within the inundated area during the 15th to 20th were partitioned for quantitative statistics to obtain specific road damage, as shown in Table 2.

$$C = (A \cup B) - (A \cap B) \tag{6}$$

where $A$ is the set of road elements within the water body before the storm, $B$ is the set of road elements within the water body after the storm, and $C$ is the set of road elements within the affected water body.

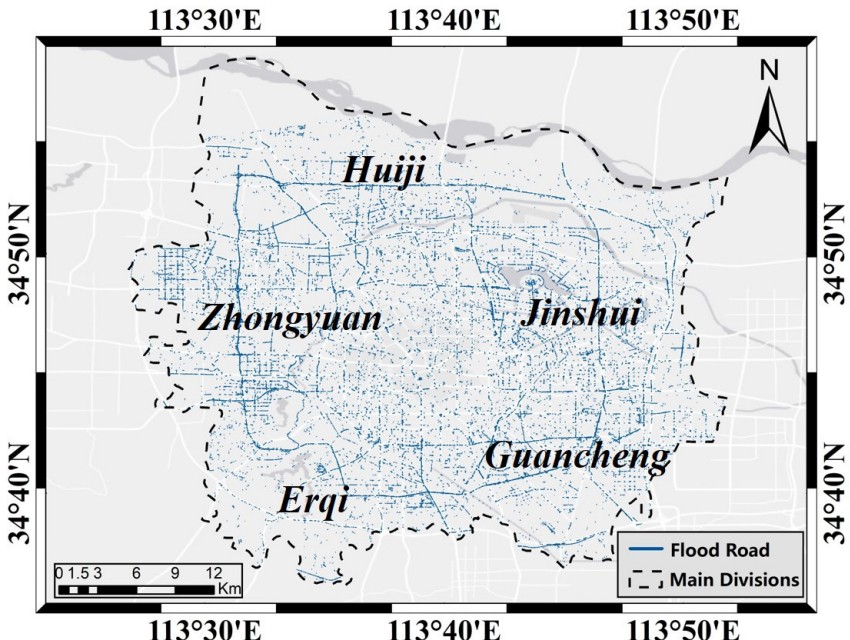

**Figure 9.** Flooded Road network in Zhengzhou.

The monitoring results in Table 2 show that the roads in the main urban area of Zhengzhou are seriously affected, with a total length of 1324.63 km. The most seriously flooded road among the five main urban areas is in Zhongyuan District, with a total inundation length of 349.17 km. The reason for such a phenomenon is that this jurisdiction is one of the earliest built-up areas in Zhengzhou. Due to the lack of scientific planning and imperfect drainage facilities in the early stages, the high-intensity rainfall exceeded its drainage capacity in a short period, resulting in many small-scale rainstorms waterlogging within a certain range. In addition, the long-term urban construction makes the residential buildings and various public facilities in this area have a high density, and the road network is relatively dense. The natural surfaces and grass are covered with asphalt or concrete, and the area of impervious water has increased dramatically. A large amount of rainwater floods the dense urban roads due to the lack of urban drainage capacity under heavy rainfall in a short period, which in turn causes numerous safety hazards.

**Table 2.** Statistics of the flooded length of the road network in each district.

| Type | Erqi | Jinshui | Zhongyuan | Huiji | Guancheng | Total |
|---|---|---|---|---|---|---|
| Length of affected road network/km | 167.94 | 327.13 | 349.17 | 189.66 | 290.73 | 1324.63 |

## 5. Discussion

### 5.1. Time Sequence of Disaster Roads in Key Areas

#### 5.1.1. Time Series Analysis of Disaster-Hit Roads in Guancheng

Guancheng District is a recent river alluvial plain area. Its terrain is generally high in the southwest and low in the northeast, with a sloping slope of about 2% and an altitude between 100 and 150 m. The main rivers in the area are the Xionger River, Qili River, and Chao River, with a total watershed area of 435.9 square kilometers, all of which belong to the Huai River system.

Guancheng District was more seriously affected by this storm disaster, and this section selects the district as the characteristic area for the time-series analysis of the flooded roads in the storm disaster. Figure 10 shows the time-series changes of the water body area in Guancheng District on 15 July, 20 July, and 22 July, and the specific water body

area of that day is indicated in Figure 10. From the water body area, Guancheng District increased rapidly on the 20th. A comparison of the time-series change map shows that the spatial distribution of the location of the increase in water body area on the 22nd is generally more dispersed. However, in the northeast of Guancheng District, Wu Zhuang Village (Figure 10 in the red circle area) shows an obvious aggregation-type increase. Wu Zhuang village is adjacent to the Dongfeng Canal to the north, the main function of which is to drain water for urban sewage. It was reported that the roads in the city were seriously waterlogged on 20 July, and the Dongfeng canal continued to operate at high water levels as an urban drainage channel. Dongfeng Canal is the main flood relief river during heavy rainfall in Zhengzhou. Xiong'er River, Qili River, and other major urban rivers converge here, and their precipitation sinks into the Jialu River, flowing from northwest to southeast to achieve the drainage effect. Hence, the water flow of the canal is large. In the vicinity of Wuzhuang Village, two rivers converge, and the place is located at the bend of the Dongfeng Canal, where the rapid flow of water can easily form on land and cause flooding.

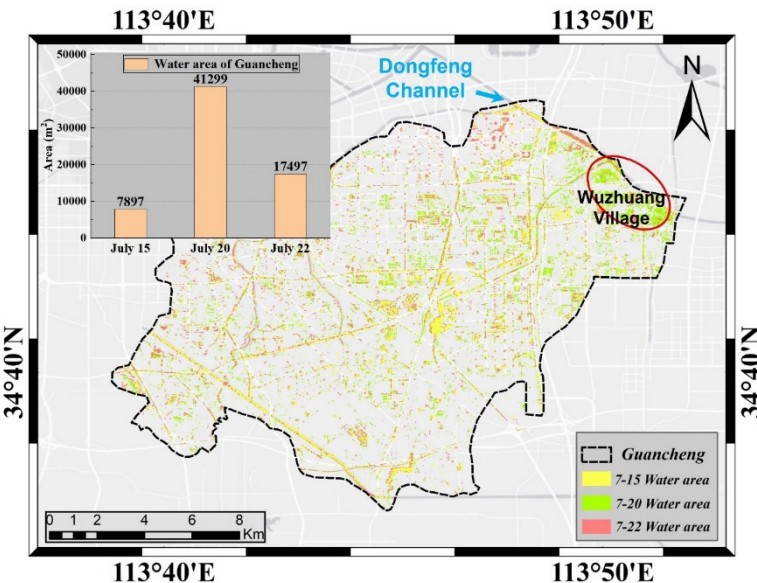

**Figure 10.** Guancheng District 15, 20, and 22 July water body area time series changes (the upper left corner of the figure shows statistics for the total area of water bodies in Guancheng District time series changes; blue arrows point to the location of the East Wind Drain; the red circle is the location of Guancheng District Wu Zhong Village).

5.1.2. Time Series Analysis of Disaster-Hit Roads in Zhongyuan

Due to long-term geological changes in the Central Plains, the southwestern part of the district is an eroded and shallow hilly area, and the rest is a loess-like inclined plain. According to the geomorphology, 78% of the region's total area is plains, and 22% is hills. The entire topography of the region is high in the west, low in the east, high in the southwest, and low in the northeast, i.e., the southwest to the northeast slope.

This section further explains the temporal change of the water body area in the Central Plains District. The temporal change of the water body area in the Central Plains District on 15, 20, and 22 July is detailed in Figure 11. Compared with 15 July, the water body area in the Central Plains District increased by 27,174 m$^2$ on 20 July. The spatial distribution of the increased water body area is more scattered, and no aggregation-type water accumulation occurred. The total reservoir capacity of Changzhuang Reservoir in Zhongyuan District is 17.4 million cubic meters, and the storage capacity of Xingli is 7.14 million cubic meters. The storage capacity of Changzhuang Reservoir effectively relieved the flood pressure in Zhongyuan District and reduced the risk of flooding.

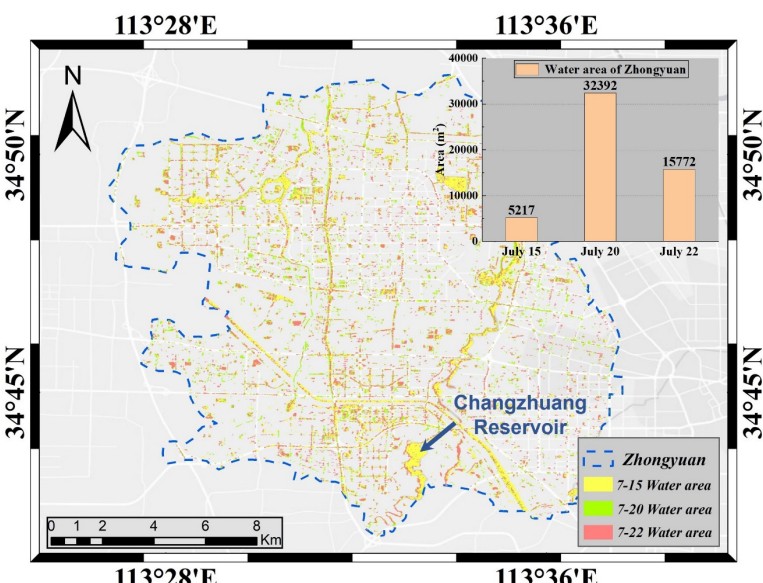

**Figure 11.** Time-series changes of water body areas in the Central Plains District on 15, 20, and 22 July (the upper right corner shows the time-series changes of total water body areas in the Central Plains District). The blue arrow is the location of Changzhuang Reservoir.

### 5.2. Limitations and Potential Improvements

According to the news released by China's Central Weather Bureau, from July 17 to 23, 2021, Henan Province suffered from severe flooding because of the historically rare and extraordinarily heavy rainfall [24]. Especially the heaviest rainfall occurred in Zhengzhou city on 20 July, and the one-hour rainfall in Zhengzhou reached 201.9 mm between 16:00 and 17:00 on that day, breaking the extreme historical value of hourly rainfall in mainland China. In this experiment, the GF-3 image acquisition time on 20 July was 22:28, which failed to monitor the maximum change in water level in time. The surface water level appeared to fall back, which in turn had some influence on the extraction of water from the inundation area and the road damage statistics in this experiment and might make the experimental results lower than the actual damage situation.

This paper draws on the threshold segmentation algorithm to extract the area of water bodies that is too coarse, and the classification accuracy still needs to be improved. In the future, we will introduce topographic features, texture features, etc., and use deep learning and other methods to improve the extraction accuracy of water bodies and obtain more accurate boundary information for inundated water bodies. At the same time, land use data is introduced to estimate the inundation of different land types, such as arable land, forest land, and residential land, to achieve rapid monitoring and post-disaster assessment of flooding.

### 6. Conclusions

To solve the problems of difficult imaging and untimely acquisition of disaster information during flooding by traditional optical remote sensing means, this paper designs a time-series monitoring scheme for flooding by combining heterogeneous SAR images with the "7-20" rainstorm in Zhengzhou City as an example. By studying two types of heterogeneous SAR data, Sentinel-1A and GF-3, which were in transit during the "7-20" rainstorm in Zhengzhou City, the scheme successfully realized the dynamic monitoring and information extraction of flooding before and after the mega rainstorm by using the threshold segmentation method. Meanwhile, remote sensing and GIS data are fused to analyze the damage to urban roads under the influence of flooding, demonstrating the capability and application of obtaining disaster spatial and temporal information from multiple perspectives supported by multi-source SAR data. The results show that: (1)

Guancheng District was the most affected area in the "7-20" rainstorm disaster in Zheng-zhou City, and the total area of water bodies in this district increased by 33,402.2 m² by 20 July 2021. The recovery of the affected area in Erqi District was slow. As of 22 July 2021, the water body area in Erqi District has been reduced by 12,461.1 m² compared with that on July 20. This phenomenon is related to the dam failure of Guojiazui Reservoir in Erqi District. (2) In this rainstorm disaster, the total length of the road network in the main urban area of Zhengzhou was affected by 1324.63 km, of which the road network in Zhongyuan District was most seriously affected, reaching 349.17 km.

This scheme adopts two kinds of heterogeneous SAR images to obtain detailed information on flood time sequence changes in the "7-20" rainstorm disaster in Zhengzhou City. It demonstrates the advantage of a joint observation scheme using heterogeneous radar image data in extracting timely and accurate information on surface flooded water bodies in rainstorms and flood disasters.

**Author Contributions:** Conceptualization, S.L.; methodology, G.J.; validation, J.L.; formal analysis, S.L.; investigation, J.L.; writing—original draft preparation, S.L.; writing—review and editing, G.J. and J.L.; visualization, J.L.; supervision, G.J.; funding acquisition, G.J. All authors have read and agreed to the published version of the manuscript.

**Funding:** This research was funded by the National Natural Science Foundation of China under grants 41474010 and 61401509, and the Natural Science Foundation of Henan Province under grant 182300410007.

**Institutional Review Board Statement:** Not applicable.

**Informed Consent Statement:** Not applicable.

**Data Availability Statement:** Sentinel-1A (https://scihub.copernicus.eu/dhus/#/home accessed on 20 December 2021), SRTM DEM (http://srtm.csi.cgiar.org/srtmdata. accessed on 12 December 2021), satellite precise orbit data (https://qc.sentinel1.eo.esa.int/aux_poeorb accessed on 20 December 2021).

**Acknowledgments:** The author would like to express my sincere thanks to Guowang Jin and Jiahao Li for their guidance on this article. This research is funded by the National Natural Science Foundation of China under Grants 41474010 and 61401509. Sincere thanks are given for the comments and contributions of anonymous reviewers and members of the editorial team.

**Conflicts of Interest:** The authors declare no conflict of interest.

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
