# Peer review of "Monitoring of “7-20” Rainstorm Damage in the Zhengzhou Road Network Using Heterogeneous SAR Images"

_applsci, doi:10.3390/app13021103_

Round 1
Reviewer 1 Report
Dear authors, thank you for your contribution.
I have several remarks or questions:
row41 , EM alghoritm - define the abbreviation
row 42 Mason et al. give a citation in bracklets, the same by other citations, after the name should be the number in bracklets (general in the introduction)
Fig.1 add a small schematic map of China, where it is located
Fig.2 improve the image, there are nothing to view
Fig. 4 part a) and b) are on first view the same, use a detail for both; if you use geocoded image, add a bar scale and orientation to the NOrth
row 208 here you describe a basic theory for SAR, this should be indicated
Fig 6 what it is Type in graph?
Fig.7 it hardly to view diferences, may be different colour (blue on gray?)
Fig.8 it is realy in squere metres (on vertical axe are hectars, 1-4, they are very small areas for flooding
May be, if possible, add an image from meteorological satellite, on this should be this storm good visible
of water bodies in this district increased by 33402.2m2 by July...really square metres? 3 hectares only? It is nothing
row 467
12461.1m2 by July 22, 2021, compared with July 20. 12,461.1m2
I think the numbers are same (wrong) and in firs case you use 12461.1, but after this 12,461.1 (join it)
row 471District was most seriously affected, reaching 349.17 km...may be square km?
General comment:it is not very clear how the damage to the road network can be detected from the data
Reference:
you use different types for names - upper case and first upper case only; it should be same, see template.
Add more references - flooding or after flooding landslides are good visible, may be add some info in itroduction, write some sentences about InSAR possibilities; you use only single images and classification.
e.g.
10.1016/j.proeps.2015.08.113, 10.5194/isprs-archives-XLI-B7-763-2016
or
https://doi.org/10.3390/rs14153718
https://www.academia.edu/43115064/SAR_and_InSAR_for_flood_monitoring_Examples_with_COSMO_SkyMed_data
or
https://www.int-arch-photogramm-remote-sens-spatial-inf-sci.net/XLIII-B3-2022/1197/2022/isprs-archives-XLIII-B3-2022-1197-2022.pdf
But it is up to you.
Author Response
Dear Reviewer,
We are very grateful for your comments on the manuscript. Some of the details the reviewer raised with us were aspects we had not considered when writing. According to your advice, we amended the relevant part in the manuscript. All of your questions were answered one by one.

Reviewer 2 Report
The paper has to be shortened, it is too long, and this discourages potential readers. It may be necessary to put some of the material in an appendix.
About half of the figures are alright but the other half have to be made larger. The figures are the source of the data and have to made larger and clearer.
Author Response
Dear Reviewer,
We are very grateful for your comments on the manuscript. Some of the details the reviewer raised with us were aspects we had not considered when writing. According to your advice, we amended the relevant part in the manuscript. All of your questions were answered one by one.Please see the document for details.

Reviewer 3 Report
1) The introductory part of the text is too much and not condensed enough. It is suggested that the introductory part be condensed again.
2) The figure name of Figure 5 is too brief and does not clearly explain the basic information shown in the figure.
3) In order to facilitate readers to make comparisons, it is suggested that the information of each district name be marked in Figures 5 and 9.
4) Some of the contents in chapter 5.2 are repeated, and it is suggested that the author reorganize this section.
5) It is suggested that the author re-check some of the punctuation and wording errors in the article again.
Author Response
Dear Reviewer,
We are very grateful for your comments on the manuscript. Some of the details the reviewer raised with us were aspects we had not considered when writing. According to your advice, we amended the relevant part in the manuscript. All of your questions were answered one by one. Please see the document for details.

Round 2
Reviewer 1 Report
Dear authors,
ok, thank you for improved text, I havent more questions.